# Early stage NSCLS patients' prognostic prediction with multi-information using transformer and graph neural network model

Jie Lian[1†], Jiajun Deng[2†], Edward S Hui[3,4], Mohamad Koohi-Moghadam[5], Yunlang She[2], Chang Chen[2*], Varut Vardhanabhuti[1*]

[1]Department of Diagnostic Radiology, Li Ka Shing Faculty of Medicine, The University of Hong Kong, Hong Kong, China; [2]Department of Thoracic Surgery, Shanghai Pulmonary Hospital, Tongji University School of Medicine, Shanghai, China; [3]Department of Imaging and Interventional Radiology, The Chinese University of Hong Kong, Hong Kong, China; [4]Department of Psychiatry, The Chinese University of Hong Kong, Hong Kong, China; [5]Division of Applied Oral Sciences and Community Dental Care, Faculty of Dentistry, The University of Hong Kong, Hong Kong, China

*For correspondence:
changchenc@tongji.edu.cn (CC);
varv@hku.hk (VV)

[†]These authors contributed equally to this work

Competing interest: The authors declare that no competing interests exist.

## Abstract

**Background:** We proposed a population graph with Transformer-generated and clinical features for the purpose of predicting overall survival (OS) and recurrence-free survival (RFS) for patients with early stage non-small cell lung carcinomas and to compare this model with traditional models.

**Methods:** The study included 1705 patients with lung cancer (stages I and II), and a public data set for external validation (n=127). We proposed a graph with edges representing non-imaging patient characteristics and nodes representing imaging tumour region characteristics generated by a pretrained Vision Transformer. The model was compared with a TNM model and a ResNet-Graph model. To evaluate the models' performance, the area under the receiver operator characteristic curve (ROC-AUC) was calculated for both OS and RFS prediction. The Kaplan–Meier method was used to generate prognostic and survival estimates for low- and high-risk groups, along with net reclassification improvement (NRI), integrated discrimination improvement (IDI), and decision curve analysis. An additional subanalysis was conducted to examine the relationship between clinical data and imaging features associated with risk prediction.

**Results:** Our model achieved AUC values of 0.785 (95% confidence interval [CI]: 0.716–0.855) and 0.695 (95% CI: 0.603–0.787) on the testing and external data sets for OS prediction, and 0.726 (95% CI: 0.653–0.800) and 0.700 (95% CI: 0.615–0.785) for RFS prediction. Additional survival analyses indicated that our model outperformed the present TNM and ResNet-Graph models in terms of net benefit for survival prediction.

**Conclusions:** Our Transformer-Graph model was effective at predicting survival in patients with early stage lung cancer, which was constructed using both imaging and non-imaging clinical features. Some high-risk patients were distinguishable by using a similarity score function defined by non-imaging characteristics such as age, gender, histology type, and tumour location, while Transformer-generated features demonstrated additional benefits for patients whose non-imaging characteristics were non-discriminatory for survival outcomes.

**Funding:** The study was supported by the National Natural Science Foundation of China (91959126, 8210071009), and Science and Technology Commission of Shanghai Municipality (20XD1403000, 21YF1438200).

### Editor's evaluation

This work constructed a population graph deep learning model using machine learning-generated imaging features and non-imaging clinical characteristics that were proven to be effective at predicting the survival of patients with early-stage NSCLC, which help us understand the imaging and non-imaging features in determining NSCLC populations with a high risk of recurrence, and the high predictive accuracy proves its novelty and significance.

## Introduction

Lung cancer is expected to account for more than 1.80 million deaths worldwide in 2021, making it the top cause of cancer-related mortality (*Siegel et al., 2021*). In early stage (stages I and II) non-small cell lung carcinomas (NSCLC), surgical resection remains the therapy of choice. However, almost 40–55% of these tumours recur following surgery (*Ambrogi et al., 2011*). The clinical care of lung cancer patients would substantially benefit from accurate prognostic evaluation. Currently, TNM staging system of lung cancer based on the anatomic extent of disease is well recognised and widely adopted, which allows tumours of comparable anatomic extent to be grouped together (*Goldstraw et al., 2016*). Staging guides treatment and provides a broad prediction of prognosis, however individual characteristics, histology, and/or therapy characteristics may impact survival results, as seen by variation within stage groups. In the refinement of the staging system, non-anatomical predictors such as gene mutations and biomarker profiles were proposed to be incorporated (*Giroux et al., 2018*). However, the gene profiling approach relies on tissue sampling, and in addition, may not fully explain the intratumoural heterogeneity seen in NSCLC. Besides, such tests have barriers in deploying to routine oncology workflows due to high turnaround time, complexity, and cost (*Malone et al., 2020*).

To predict the patient's prognosis and to optimise individual clinical management, prognostic predictors such as TNM system and imaging-based high throughput quantitative biomarkers, radiomics, have been widely used to describe tumours (*Du et al., 2019*; *Aonpong et al., 2020*; *Bera et al., 2022*; *Carmody et al., 1980*; *Chirra et al., 2019*; *Mirsadraee et al., 2012*; *van Griethuysen et al., 2017*). Artificial intelligence (AI) methods, especially some deep learning (DL) models, have recently been regarded as potentially valuable tools (*Chilamkurthy et al., 2018*; *Nabulsi et al., 2021*; *Xu et al., 2019*). DL models generated multiple quantitative assessments for tumour characteristics, which have the potential to describe tumour phenotypes with more predictive power than the clinical model (*Xu et al., 2019*). While the anatomical structures in a medical image are functionally and mechanically related, most AI-based methods do not take these interdependencies and relationships into account. This leads to instability and poor generalisation of performance (*Zhou et al., 2021*). With recent advancements in AI technology, several novel models have been proposed. Notably, the Transformer (*Vaswani et al., 2017*) model permits exceptional capabilities in natural language processing fields such as language translation and was later applied to the computer vision field and outperformed all state-of-the-art models given large amounts of training data (*Dosovitskiy et al., 2020*). This provides an intuitive reason to apply the Transformer model to the medical image to generate additional meaning for tumour features, as images were processed in sequence with inherent interdependencies (*Zhou et al., 2022*).

The majority of current prognostic prediction methods have focused mainly either specific to their own domains, such as focusing solely on imaging data, whereas in clinical practice non-imaging clinical data such as sex, age, and disease history all play critical roles in disease prognosis prediction (*Holzinger et al., 2019*). Although some researchers have used multi-modal techniques (*Xue et al., 2018*) to combine that information, it is not easy to explain how the various types of data interacted and how they contributed to the final prediction. Due to their lack of explanatory power, those models may not be easily applied in clinical practice (*London, 2019*). Another type of neural network, called a graph neural network (GNN) (*Kipf and Welling, 2016*), which deals with data that has a graph structure, enables researchers to create more flexible ways to embed various types of data. For example, nodes and edges in a graph might represent a variety of different types of data (imaging and clinical demographics information), and analysing these entities reveals the role of various data sources.

In this study, we proposed a GNN-based model that leverages imaging and non-imaging data for the prediction of the survival of patients with early stage NSCLC. Patients were represented as a population graph, whereby each patient corresponded to a graph node and was associated with a

tumour feature vector that was learnt from the Transformer model, and graph edge weights between patients were derived from a similarity score that was derived from phenotypic data, such as demographics, tumour location, cancer type, and TNM staging. This population graph was used to train a GraphSAGE (*Hamilton et al., 2017*) model for classifying individual patient's risk of overall survival (OS) and recurrence. Additionally, we attempted to determine the relative importance of imaging and non-imaging features within this model. The proposed model was trained and tested on a large data set, followed by external validation using a publicly available data set.

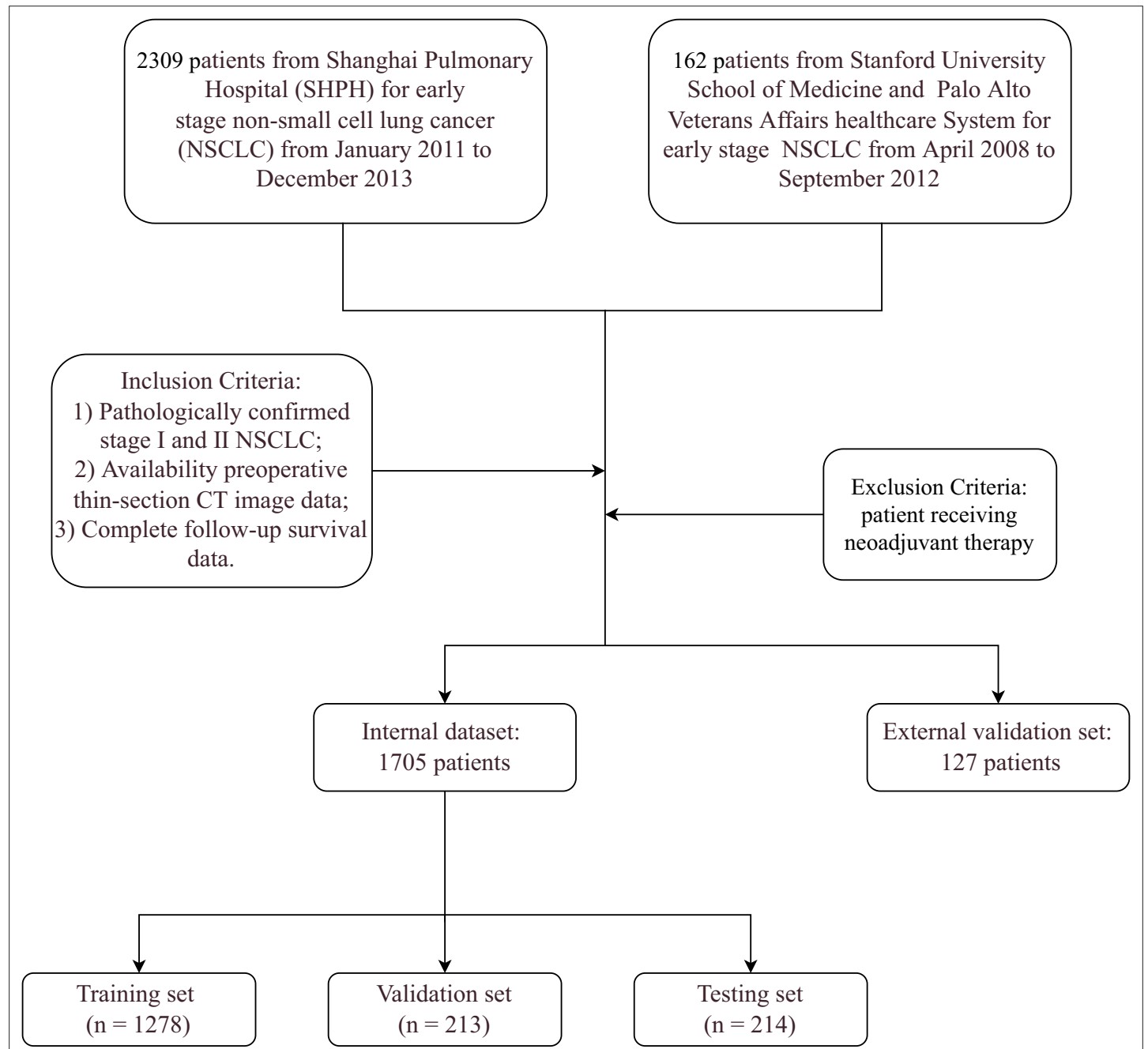

**Figure 1.** Overall flow of the study in both internal and external data set.

## Methods

### Participants

The study included consecutive patients who received surgery for early stage NSCLC between January 2011 and December 2013 who matched the criteria. Inclusion criteria included: (1) pathologically proven stage I or stage II NSCLC; (2) preoperative thin-section CT image data; and (3) complete follow-up survival data. Patients undergoing neoadjuvant therapy were excluded from the study. This retrospective study protocol was approved by the Shanghai Pulmonary Hospital's Institutional Review Board (ref: L21-022-2) and informed consent was waived owing to retrospective nature. Additionally, patients who met our criteria were retrieved from the NSCLC Radiogenomics (*Bakr et al., 2018*) data set as an external validation set (see *Figure 1* for the internal and external inclusion criteria flowchart).

We only used patients' initial CT scans in this study. For the main cohort, all CT scans were acquired using Somatom Definition AS+ (Siemens Medical Systems, Germany) and iCT256 (Siemens Medical Systems, Germany; Philips Medical Systems, Netherlands). All image data were rebuilt using a 1-mm slice thickness and a 512×512 mm$^2$ matrix. Intravenous contrast was administered in accordance with institutional clinical practice. Clinical data in this study were manually collected from medical records and were anonymised. Outpatient records and telephone interviews were used to collect follow-up data. The period between the date of surgery and the date of death or the final follow-up was defined as OS. Recurrence-free survival (RFS) was calculated from the date of surgery to the date of recurrence,

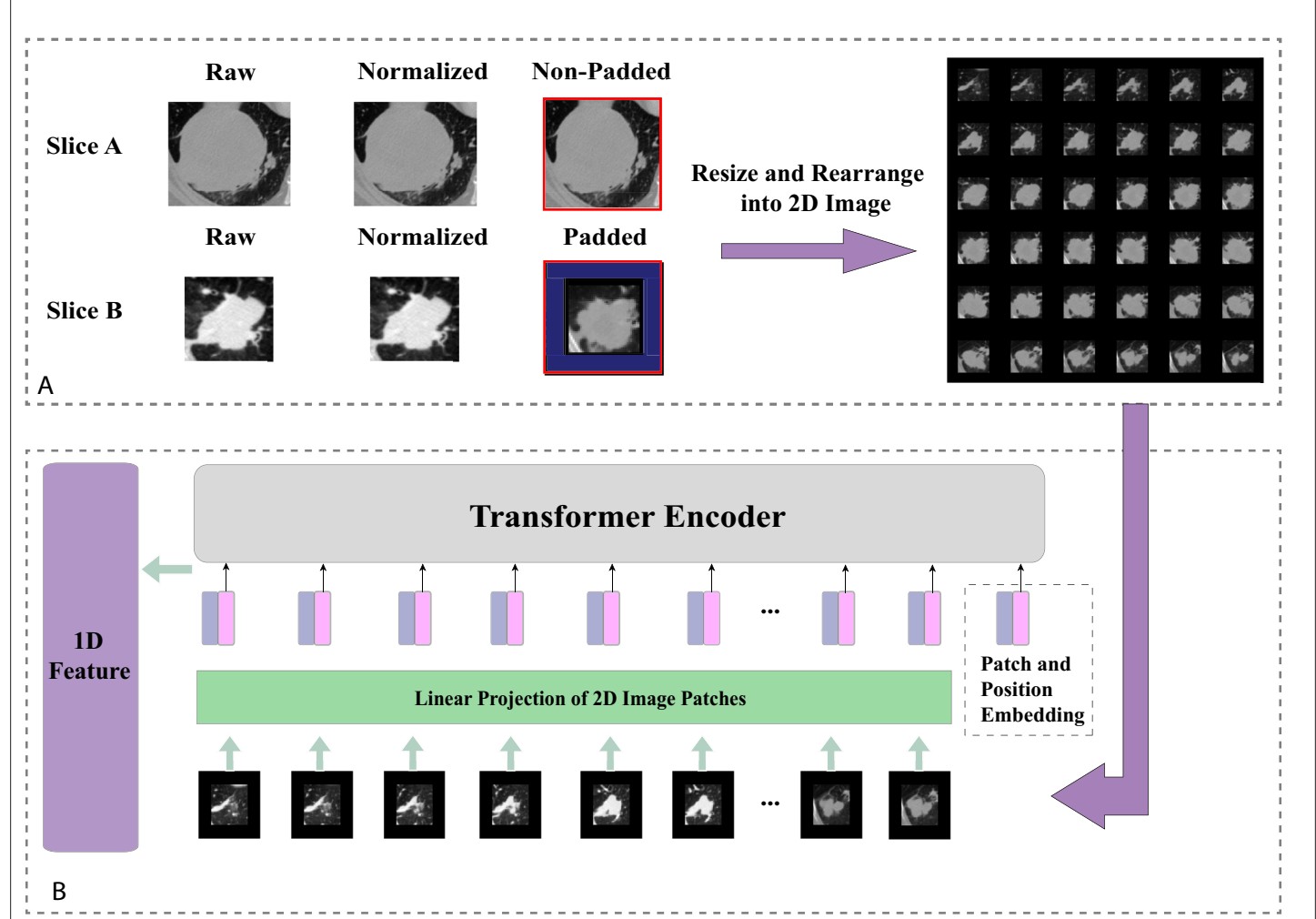

**Figure 2.** Tumour image processing and feature generation. (**A**) Tumour images normalisation, reshaping, and padding to standard sizes, then rearranged into 2D images. (**B**) Generating 1D Transformer survival features from pretrained Transformer model.

death, or last follow-up. (More details about internal scan parameters and follow-up strategies can be found in Appendix 1).

## Image annotation and pre-processing

Patients' tumour region was manually labelled by experienced radiologists using 3D Slicer (*Fedorov et al., 2012*), with a centre seed point defining a bounding box. The regions of interest (ROIs) were first annotated by two junior thoracic surgeons (Y.S. and J.D. with 5 and 3 years of experience, respectively), then the consensus on ROI was obtained by a discussion with a senior radiologist (with more than 25 years of experience).

For image pre-processing, we first normalised all CT images and removed the surrounding noises such as bones by manual thresholding. The size of all tumour segments was fixed to 128×128×64 mm$^3$. Small tumours were zero-padded. To reduce the computational cost, we resized the padded segments into 64×64×36 mm$^3$ and subsequently resized them as 2D square images (each row contained six tumour slices) with the size of 384×384 mm$^2$ as shown in *Figure 2A*.

## Tumour Transformer feature generator

When pretrained on a large data set and transferred to image recognition benchmarks, it has been shown that Vision Transformer (ViT) can achieve excellent results while requiring significantly less computational resources to train than state-of-the-art convolutional models (*Dosovitskiy et al., 2020*). To this end, we reasoned that by replacing the traditional CNN feature generator architecture with a Transformer structure could be an approach to produce meaningful survival-relevant features. In this study, we used a ViT pretrained on a large-scale data set (ImageNet-21k; *Ridnik et al., 2021*) as the feature generator, which takes 2D tumour segments as inputs. To meet the standard requirements of the sequence model, the input images were divided into 36 ordered patches and position embedding in the first step, followed by a linear projection function before entering the Transformer Encoder. We replaced the original classification layer with a fully connected layer to generate a 1D feature vector. The detailed implementation is illustrated in *Figure 2B*. The 1D feature vector was then assigned as the node feature for the individual patient in the graph network.

## Patient survival graph network

A population graph method was used to leverage imaging and non-imaging data. Each patient was regarded as a node in a graph and its edge with neighbour was derived from a similarity score which was determined by the product between four component scores, namely demographics (gender and age), tumour location, cancer type (histology), and TNM staging (for more detail, refer to the Appendix 1 for a detailed explanation of similarity scores). Two patients would be connected to each other if they shared similar component scores. The features of an individual patient (node feature) were obtained from the Transformer Encoder trained on the tumour images mentioned above.

## Graph-based neural network structure

We applied a graph-based deep neural network structure called GraphSAGE in this study. The proposed network took the whole population graph, along with the edge and node features as the input and generated a risk score in the last layer for each patient node as the output (see *Figure 3*). Within the network, every node feature was updated by an aggregation of information from its neighbours and itself, while the importance of different neighbours varied by the corresponding edges' weight.

We applied a two-layer GraphSAGE and global meaning pooling structure, aiming to allow each patient's information to be updated, first from its second neighbours and then its neighbours and itself consequentially. In order to emphasise the target of survival prediction, we specifically replaced the cross-entropy loss with Cox proportional hazards loss function (*Katzman et al., 2018*) which both considered the survival time and events when training the network. The proposed network was implemented in Python, using the Deep Graph Library with Pytorch backend.

## Statistics analysis

All patients from the main data set were randomly separated into training, validation, and testing sets with the proportion of 75%, 12.5%, and 12.5% separately. We also tested the model on the external

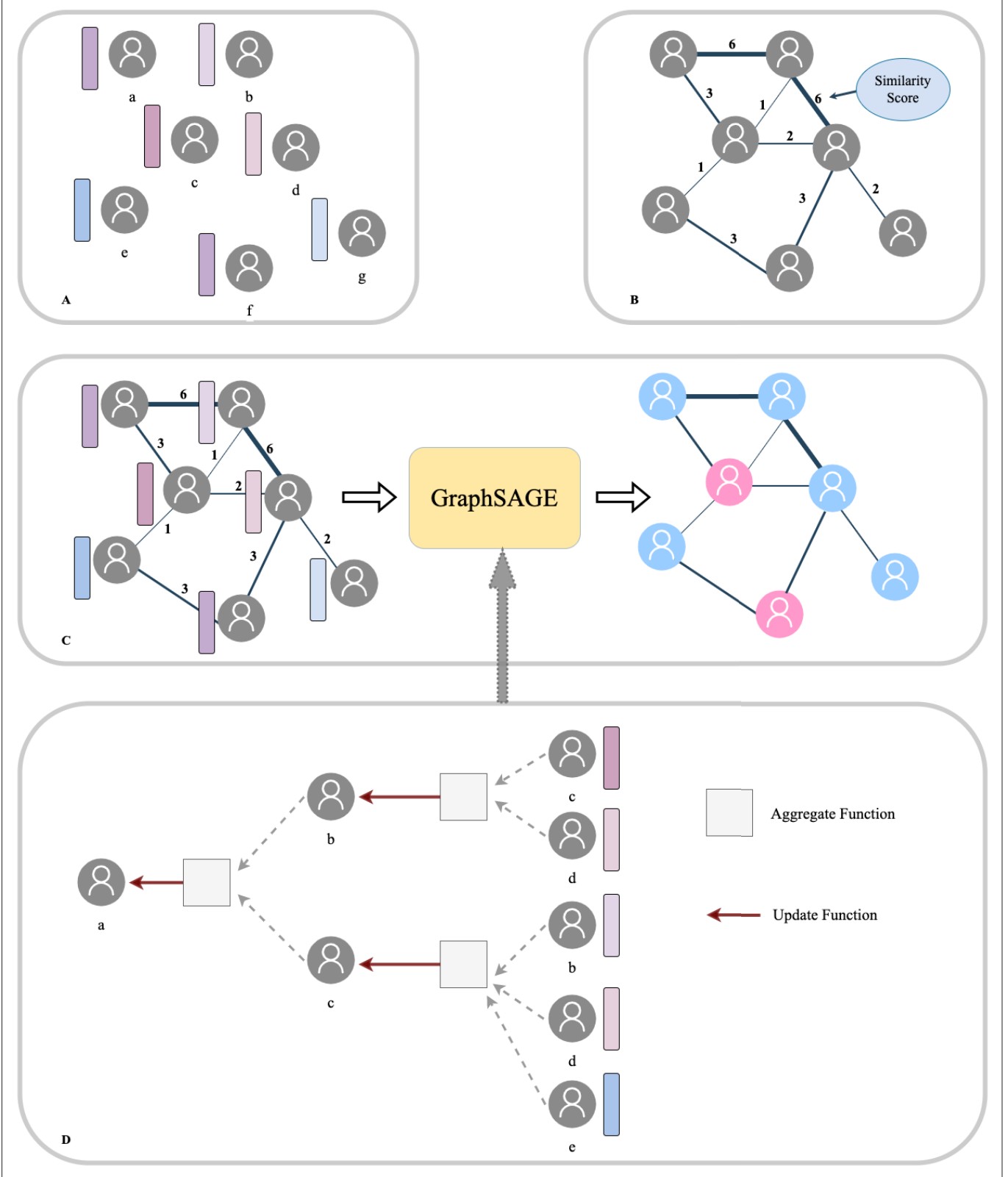

**Figure 3.** Population graph building and model prediction pipeline. (**A**) Each patient was regarded as a node and the Transformer-generated feature was regarded as node features. (**B**) Graph edges and the relevant weights were defined by their similarity scores. (**C**) We then put the whole population graph to train the GraphSAGE network in order to make a prediction for each patient (pink indicates high risk and blue indicates low risk). (**D**) Node updating inside the GraphSAGE network.

validation data set. The proposed model was compared with the TNM staging system which was generally used in clinical practice and a ResNet-Graph model which has the same graph structure (using clinical features to define the similarity score) as our proposed model while the node feature was generated by a pretrained ResNet-18 model (*Khanna et al., 2020*; *Chen et al., 2019*) (using imaging features).

To evaluate whether there were statistically significant variations in survival between positive and negative groups, the area under the receiver operator characteristic curve (AUC) was determined for OS and RFS prediction to compare the models' performance. The Kaplan–Meier (KM) method was used to generate prognostic and survival estimates for groups with low and high risk (both for OS and RFS), which were stratified according to the training set's median prediction probability, with the log-rank test employed to establish statistical significance. To quantify the net benefits of survival prediction, we quantified the net reclassification improvement (NRI) and integrated discrimination improvement (IDI), as well as performed a decision curve analysis (DCA). All of the analyses above were performed in Python using the Lifelines package.

An additional subanalysis was performed on the test data set to explore the relationship between patients' clinical information and imaging features contributing to risk prediction. We generated a sub-graph visualisation using PyVis and a KM analysis was used for several sub-graphs to evaluate our model's ability to separate high-risk patients. Finally, as a proof of concept, we plotted two patients' nodes feature changes before and after one layer processing using a correlation heatmap, along with its neighbours' edge weights analysis to try to understand the inner workings of our model.

## Results

### Data description

In the main cohort, we initially enrolled 2309 patients and after exclusion based on our criteria, a total of 1705 NSCLC patients were included in the study. The median age was 61 (interquartile range, 55–66 years). There were 1010 males (59.2%) and 695 women (40.8%). Tumours were more frequently located in the upper lobes (1018, 59.7%). A total of 1235 patients (72.4%) had adenocarcinoma, while 391 patients (22.9%) had squamous cell carcinoma. The distribution of pathologic stages was as follows: stage IA was present in 791 patients (46.4%), stage IB was present in 607 patients (35.6%), stage IIA was present in 133 patients (7.8%), and stage IIB was present in 174 patients (10.2%). The OS and RFS rates were 78·2% (95% confidence interval [CI]: 76.2–80.2%) and 74.2% (70.8–77.6%), respectively. The external validation data set included a total of 127 patients of which 32 (25.2%) were females and 95 (74.8%) males, with a median age of 69 (interquartile range, 46–87 years). Upper lobe tumours were also more prevalent (76 patients, 59.8%). Among them 95 patients were diagnosed with adenocarcinoma and 30 with squamous cell carcinoma. The OS and RFS rates were 68.5% (95% CI: 60.4–77.7 %) and 59·1% (95% CI: 50.4–67.8 %), respectively. Please refer to *Table 1* for more detailed information.

### Model performance

To develop deep transformer graph learning-based biomarkers for OS prediction, we trained on the main cohorts, separated into training and validation data sets and then evaluated them separately on the testing set (213 patients) and the external set (127 patients). For OS prediction, our model achieved AUC values of 0.785 (95% CI: 0.716–0.855) and 0.695 (95% CI: 0.603–0.787) on the testing and external data sets, respectively, compared to 0.690 (95% CI: 0.600–0.780) and 0.634 (95% CI: 0.544–0.724) for the TNM model, and 0.730 (95% CI: 0.640–0.820) and 0.626 (95% CI: 0.530–0.722) for ResNet-Graph model. For RFS prediction, our model achieved AUC values of 0.726 (95% CI: 0.653–0.800) and 0.700 (95% CI: 0.615–0.785) on the testing and external data sets, respectively, compared to 0.628 (95% CI: 0.542–0.713) and 0.650 (95% CI: 0.561–0.732) for the TNM model, and 0.681 (95% CI: 0.598–0.764) and 0.595 (95% CI: 0.615–0.785) for ResNet-Graph model (*Figure 4A and B*).

Additional survival analyses were performed using KM estimates for groups with low and high risk of mortality and recurrence, respectively, based on the median stratification of patient prediction scores (*Figure 4C and D*). All three models showed statistically significant differences in 5-year OS. For RFS prediction, the ResNet-Graph model was unable to distinguish between individuals at low

**Table 1.** Feature distribution in the total patient cohorts, training and validation cohorts and the test cohorts.

| Feature | Content | TRAIN and VAL (n=1492) | TEST (n=213) | | EXTERNAL (n=127) | |
|---|---|---|---|---|---|---|
| | | Mean, SD, 95% CI/Count, % | Mean, SD, 95% CI/Count, % | p | Mean, SD, 95% CI/Count, % | p |
| Age | Age | 60.6, 8.7, (CI: 60.1, 61.0) | 60.7, 9.5, (CI: 59.4, 62.0) | >0.05 | 68.7, 9.1, (CI: 67.2, 70.1) | <0.01** |
| Sex | Female no. (%); Male no. (%) | 602 (33.3); 890 (66.7) | 93 (33.3); 120 (66.7) | >0.05 | 32 (25.2); 95 (74.8) | <0.01** |
| Resection | Sublobar resection no. (%); Lobectomy no. (%); Bilobectomy no. (%); Pneumonectomy no. (%) | 123 (8.2); 1292 (86.6); 59 (3.95); 18 (1.2) | 23 (10.8); 180 (84.5); 7 (3.3); 3 (1.4) | >0.05 | / | / |
| Histology | Adenocarcinoma no. (%); Squamous Cell Carcinoma no. (%); Others no. (%) | 1072 (71.4); 351 (23.5); 69 (4.6) | 163 (76.5); 40 (18.8); 10 (4.7) | >0.05 | 95 (74.8); 30 (23.6); 2 (1.6) | >0.05 |
| Tumour location | LUL no. (%); LLL no. (%); RUL no. (%); RML no. (%); RLL no. (%) | 384 (25.7); 211 (14.1); 504 (33.8); 146 (9.8); 247 (16.6) | 51 (23.9); 37 (17.4); 79 (37.1); 15 (7.0) 31 (14.6) | >0.05 | 30 (23.6); 22 (17.3); 46 (36.2); 15 (11.8); 14 (11.0). | >0.05 |
| Tumour size | Tumour size | 2.68, 1.38, (CI: 2.61, 2.75) | 2.55, 1.25, (CI: 2.38, 2.71) | >0.05 | / | / |
| pTNM stage | Stage I no. (%); Stage II no. (%); | 1219 (81.7); 273 (18.3) | 179 (84.0); 34 (16.0) | >0.05 | 97 (76.3); 30 (23.7) | <0.01** |
| RFS status | RFS no. (%) | 1089 (73.0) | 154 (72.3) | >0.05 | 75 (59.1) | >0.05 |
| RFS month | RFS month | 57.5, 24.5, (CI: 56.2, 58.7) | 58.4, 23.4, (CI: 55.2, 61.5) | >0.05 | 39.5, 26.9, (CI: 34.8, 44.2) | <0.01** |
| OS status | OS no. (survival %) | 1166 (78.2) | 167 (78.4) | >0.05 | 87 (68.5) | >0.05 |
| OS month | OS month | 62.4, 19.9, (CI: 61.4, 63.4) | 63.4, 18.4, (CI: 60.9, 65.9) | >0.05 | 44.8, 27.8, (CI: 40.9, 50.0) | <0.01** |

and high risk (p>0.05), while both Transformer-Graph and TNM models were able to separate high and low risk of RFS groups (p<0.05). The KM plots for the external set were reported in *Figure 4—figure supplement 1*.

Additionally, the DCA (*Figure 4E*) and net benefit analysis (IDI, NRI) indicated that the Transformer-Graph model significantly outperformed the present TNM and ResNet-Graph models in terms of net benefit for both OS and RFS survival prediction. As for detailed net benefit analysis, Transformer-Graph model outperformed the present TNM and ResNet-Graph models in terms of IDI and NRI. Our proposed model improved the survival prediction significantly compared with TNM regarding NRI (OS: 0.284, 95% CI: –0.112 to 0.519, p<0.0001; RFS: 0.175, 95% CI: –0.115 to 0.486, p<0.0001) and IDI (OS: 0.159, 95% CI: 0.103–0.214, p=0.00032; RFS: 0.137, 95% CI: 0.086–0.189, p=0.00074). The results comparing with ResNet-Graph were reported in Appendix 1.

## Patients' clinical-based graph analysis

We visualised the whole internal set (*Figure 5A*) along with the testing cohort's subplot (*Figure 5B*) and analysed two challenging cases to better understand the population-based graph structure and how clinical data was integrated with node attributes (i.e., patients' tumour images). The testing subplot showed that while the graph structure (specified by the similarity score) was capable of broadly separating at-risk patients, several clusters had both high- and low-risk patients intermingled

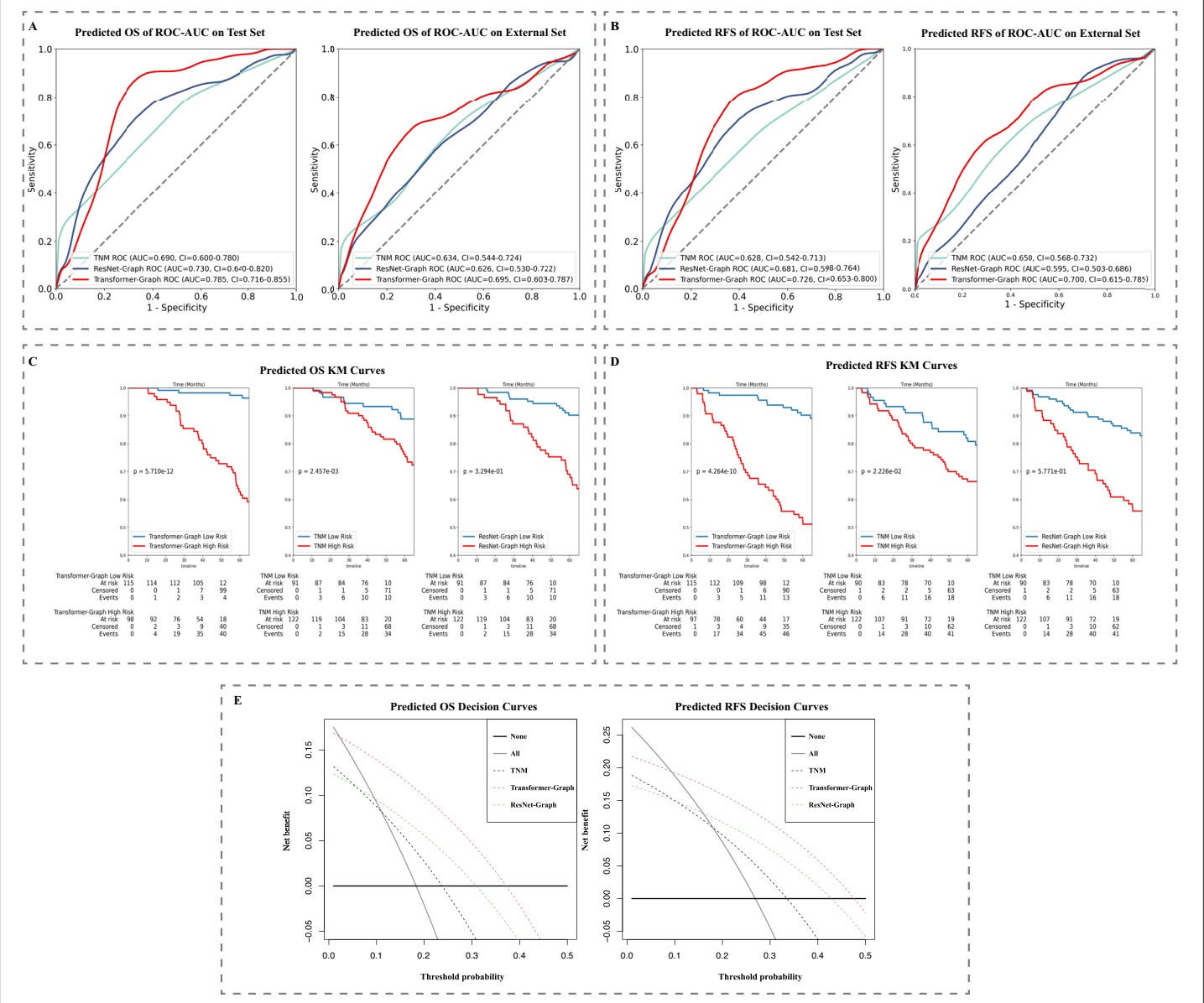

**Figure 4.** Model performance: (**A**) ROC-AUC curve on test data and external set for OS and (**B**) RFS prediction and (**C**) KM curve on test data set for OS and (**D**) RFS prediction. (**E**) Decision curve on test data set for OS and RFS prediction. KM, Kaplan–Meier; OS, overall survival; RFS, recurrence-free survival; ROC-AUC, area under the receiver operator characteristic curve.

The online version of this article includes the following figure supplement(s) for figure 4:

**Figure supplement 1.** Kaplan-Meier survival analysis.

together, making them difficult to separate using traditional clinical information (see *Figures 5C and 4D*). The subsequent KM analysis indicated that by using Transformer-generated tumour attributes, high- and low-risk patients could be significantly discriminated.

Additionally, we analysed specifically as an example, patient No. 44 (high-risk node), and surrounding neighbours' edge weights distribution, as well as the initial and subsequent one layer node features. This patient was a high-risk patient who died after 38 months, with 42 neighbours. Initially, we analysed the correlation coefficient between neighbours' node features in order to determine the role that Transformer-generated image features played prior to graph training. As illustrated in *Figure 5E*, the correlation matrix of Transformer-generated features revealed that almost all of patient No. 44's high-risk (dashed box nodes) and low-risk neighbours were highly correlated, implying that image features

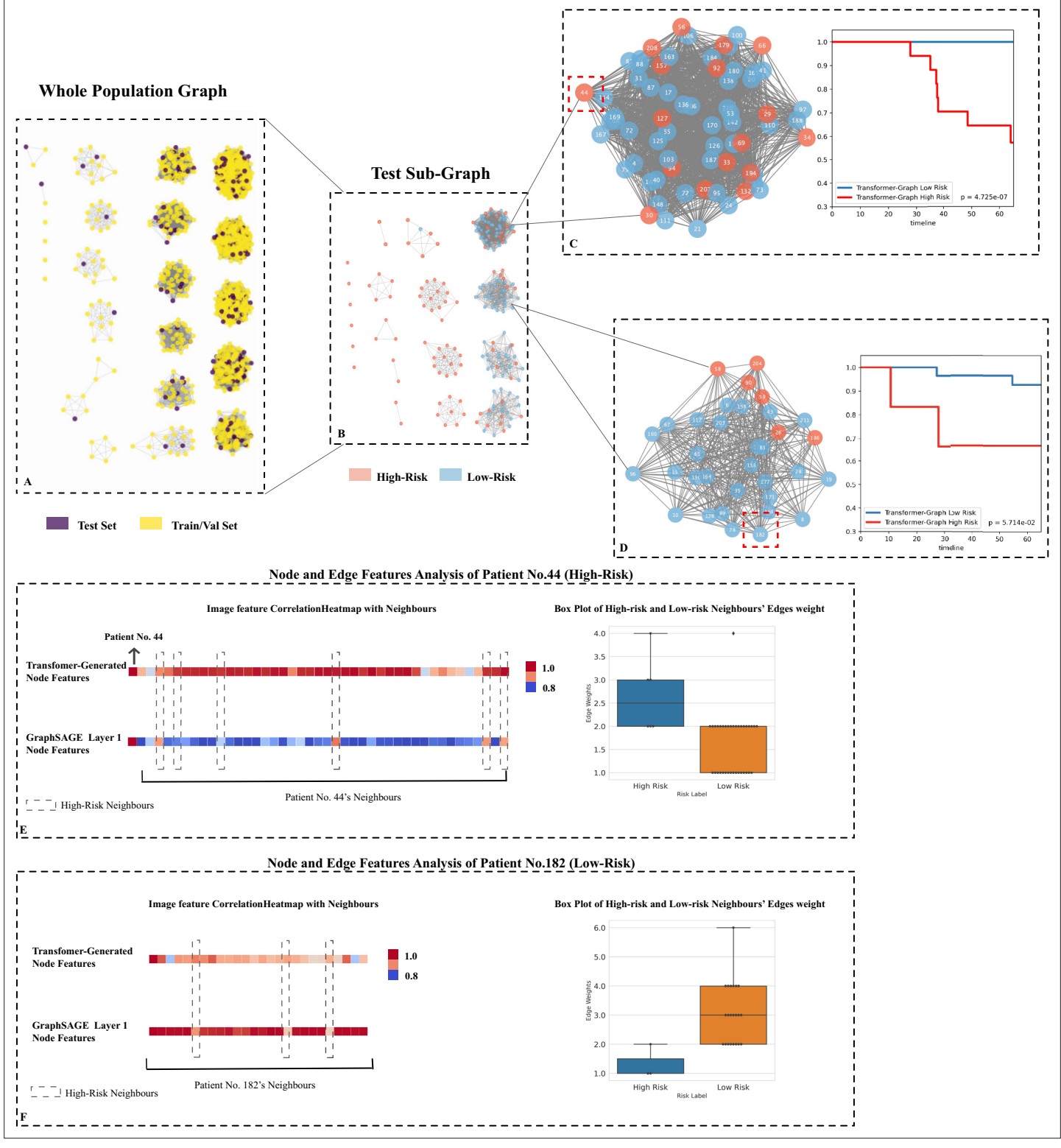

**Figure 5.** Testing set graph analysis. (**A**) A visual representation of the whole cohort population graph of 1705 patients. (**B**) A visual representation of the testing sub-graph of 213 patients. (**C**) and (**D**) two sub-graphs containing challenging cases where the graphs contained both high- and low-risk patients. (**E**) Node features' correlation heatmaps and edge weights distribution of patient No. 44: Each square represents a neighbour's node features' correlation coefficient, higher values (red colour) reveal closer relation with the target node; the box plot of 42 neighbours indicates that the high-risk neighbours (blue box) have higher edge weights median. (**F**) Node features' correlation heatmaps and edge weights distribution of patient No. 182: The box plot of 25 neighbours indicates that the low-risk neighbours (orange box) have higher edge weights median.

did not contain directly discriminative survival information before learning. We next then examined the distribution of neighbours' edge weights. As illustrated in *Figure 5E*, despite the fact that there were only five high-risk neighbours, the median value of similarity scores was slightly higher than that of low-risk neighbours (2.50 vs. 2.00), indicating that the high-risk neighbour group was more closely connected to the target nodes from non-imaging information aspects. After one layer of GraphSAGE updating, we discovered that the high-risk neighbours were more correlated with patient No. 44 (see *Figure 5E* GraphSAGE Layer 1, nodes in the dash boxes showed higher coefficient values), revealing that within our model, both neighbours' nodes and edge features contained survival-related information, and they contributed together to efficiently provide information for the target node learning.

Then, we analysed the correlation between patient No. 182 (a patient with low risk) and its 25 neighbours (see *Figure 5F*). The majority of patient No. 182's neighbours were low-risk patients (22 out of 25), and the median value of their similarity scores was significantly higher than that of their high-risk neighbours (3.00 vs. 1.00). With the help of edge weights (clinical data) updated on node features, high-risk neighbours could be separated after only one layer (see *Figure 5*, three nodes in GraphSAGE Layer 1's dashed boxes had lower coefficient values).

## Discussion

The patient's prognosis is relevant for both treatment estimation and future treatment planning. In practice, clinical information and some hand-crafted medical imaging features were used to predict outcomes (*Du et al., 2019*). With the advent of AI technologies, methods that incorporate deep-learning-based features have been developed, generating more medical imaging-related features from a variety of perspectives (*Xu et al., 2019*), resulting in improving the prediction performance. In previous studies (*Aonpong et al., 2020*; *Nabulsi et al., 2021*; *Xu et al., 2019*; *Wang et al., 2018*), convolutional models such as ResNet were commonly used, whereas ViT, which outperformed novel convolutional-based DL models in computer vision tasks on nature image data sets, could generate medical imaging features in a different manner. Besides, the design of the combination of imaging and non-imaging data is always a challenge. In the past, linear models were commonly used (*Liao et al., 2019*), treating imaging and non-imaging equally, which may have resulted in inefficient information utilization. The proposal of multimodal data integration allows for effective information fusion over different modalities (*Brown et al., 2018*), while the relation between modalities is not well-explained.

In this project, we demonstrated the feasibility of using ViT on CT images of lung tumours to generate features for cancer survival analysis. Additionally, we used a graph structure to embed patients' imaging and non-imaging clinical data separately in the GNN and attempted to explain how clinical data communicates with Transformer-generated imaging features for survival analysis. While Transformer and GNN models have been widely used in computer vision, their application in the medical field, particularly for survival prediction, is still evolving due to the complexity and unbalanced nature of medical data (high dimension, multiple data formats, including non-imaging data). In our study, we combined these two methods and created a specially designed graph structure to handle a variety of data formats, demonstrating the utility of Transformer-generated features in survival analysis and emphasising the extent to which clinical data and imaging features contribute to the prediction. To our knowledge, this is the first work to demonstrate the feasibility of using Transformer in survival prediction using a graph data structure and exploratory analysis of the models' intuitions in an attempt to explain these state-of-the-art methods.

Our experiments indicated that the proposed model outperformed the commonly used TNM model in predicting survival not only on the testing data set but also on the external dataset, despite the fact that the data distributions were significantly different (refer to *Table 1*, the survival distribution on the external data set is significantly different from the internal data set), demonstrating the model's generalisability for unseen data. The model also outperformed the generally regarded current state-of-the-art model, the Res-Net model which in our study incorporated both imaging and non-imaging data when we performed survival analyses based on KM estimates. The model's good performance indicated that both the Transformer-generated imaging features and the structure of our population graph (i.e., using graph edges and nodes to combine non-imaging clinical data and imaging data) contained useful information for survival. Additionally, the subplot graph on the testing data set (*Figure 5B*) indicated that our graph structure was capable of approximate clustering high- and low-risk groups and segregating the majority of the high-risk patients. Meanwhile, when patients

were similar in terms of demographic information and it was hard to determine the risk patients by traditional clinical methods (refer to *Figure 5C and D* the dense graphs containing both pink and blue nodes), the Transformer-generated image features and edge weights had more roles to play in determining the differences between neighbours. More specifically, the Transformer-generated features did not contain directly discriminative survival information before learning, while with edge weights together, effective information from neighbours' node features could be determined. In this case, all patient node features could be effectively updated, and high-risk patients could be better discriminated as in *Figure 5E*.

Our study contains several strengths. First, our data set is relatively large, encompassing both contrast and non-contrast CT scans. This not only aided in the model's generalisation learning but also allows for flexibility in the imaging standards in clinical settings. Second, our graph model demonstrated the ability to combine non-imaging clinical features with imaging features in an understandable manner, implying a new direction of embedding multi-data with DL models. Finally, we sought to understand the roles of imaging and non-imaging features in determining high-risk nodes within the GNN, which could aid clinicians in comprehending the internal workings of the neural networks.

There are some limitations worth noting. First, whilst the proposed model significantly outperformed the TNM model on the external data set (OS prediction AUC 0.693 vs. 0.633, RFS prediction RFS 0.700 vs. 0.650), the model's performance on the external set was below that of the testing set (AUC 0.783 and 0.726 for OS and RFS). One reason could be that the patients' demographics were different, particularly in terms of age (the external group's average age was 10 years older than the main cohort), cancer staging (84.0% stage I in the main cohort while 76.3% in the external testing set), and gender (male percentage 66.7% vs. 74.8%). The fact that the two data sets originate from distinct countries, as well as the differences in ethnicity, treatment, and follow-up strategies (see *Table 1*, especially the mean follow-up time) may also have an impact on the prediction performance (see *Supplementary file 1* for ethnicity and smoking information of external set). Second, smoking history played important role in the development of lung cancer, which may help to improve the model's performance and reduce the performance difference between the testing set and the external set, despite the fact that relevant information was not collected in the main set. Besides, we only found one applicable public external in this project, whereas more external data can improve the convince of our model's generalization ability. (We tried to search on the Cancer Imaging Archive, and only 2 of 40 lung cancer data sets meet our requirements. We used the first as our current external set; the second has a death rate of 95.88% for early stage lung cancer patients with no explanation, so we did not include it.) Finally, the initial step requires the human observer to identify the tumour and draw a bounding box which in our study was still a manual procedure. As the pipeline for automatic tumour detection and segmentation becomes more mature, this step can potentially be automated allowing for ease of translation into the clinics.

In conclusion, the population graph DL model constructed using Transformer-generated imaging and non-imaging clinical features was proven to be effective at predicting survival in patients with early stage lung cancer. The subanalysis concluded that by developing a meaningful similarity score function and comparing patients' non-imaging characteristics such as age, gender, histology type, and tumour location, the majority of high-risk patients can already be separated. Additionally, when high- and low-risk patients shared very similar demographic information, TNM information provided additional information for survival prediction when combined with tumour imaging features.

## Additional information

### Funding

| Funder | Grant reference number | Author |
| --- | --- | --- |
| National Natural Science Foundation of China | 91959126 | Chang Chen |
| National Natural Science Foundation of China | 8210071009 | Yunlang She |

| Funder | Grant reference number | Author |
| --- | --- | --- |
| Science and Technology Commission of Shanghai Municipality | 20XD1403000 | Chang Chen |
| Science and Technology Commission of Shanghai Municipality | 21YF1438200 | Yunlang She |

The funders had no role in study design, data collection and interpretation, or the decision to submit the work for publication.

## Author contributions

Jie Lian, Conceptualization, Data curation, Software, Formal analysis, Validation, Visualization, Methodology, Writing – original draft, Writing – review and editing; Jiajun Deng, Conceptualization, Data curation, Formal analysis, Validation, Visualization, Methodology, Writing – original draft, Writing – review and editing; Edward S Hui, Mohamad Koohi-Moghadam, Conceptualization, Software, Methodology, Writing – review and editing; Yunlang She, Resources, Data curation, Writing – review and editing; Chang Chen, Conceptualization, Data curation, Supervision, Methodology, Writing – original draft, Writing – review and editing; Varut Vardhanabhuti, Conceptualization, Data curation, Formal analysis, Supervision, Methodology, Writing – original draft, Project administration, Writing – review and editing

## Author ORCIDs

Jie Lian (ID) http://orcid.org/0000-0003-2351-2570
Jiajun Deng (ID) http://orcid.org/0000-0002-6834-0322
Edward S Hui (ID) http://orcid.org/0000-0002-1761-0169
Mohamad Koohi-Moghadam (ID) http://orcid.org/0000-0002-7286-0427
Yunlang She (ID) http://orcid.org/0000-0001-7673-9846
Varut Vardhanabhuti (ID) http://orcid.org/0000-0001-6677-3194

## Ethics

Human subjects: This retrospective study protocol was approved by the Shanghai Pulmonary Hospital's Institutional Review Board (ref: L21-022-2) and informed consent was waived owing to retrospective nature.

## Decision letter and Author response

Decision letter https://doi.org/10.7554/eLife.80547.sa1
Author response https://doi.org/10.7554/eLife.80547.sa2

# Additional files

## Supplementary files

• Supplementary file 1. Ethnicity and Smoking information table of the external dataset.
• MDAR checklist

## Data availability

The current manuscript is a computational study, so no data have been generated for this manuscript. To aid reproducibility of research, our codes are published on the Github repository: https://github.com/Serene-Lian/TransGNN-Lung Copy archived at swh:1:rev:a8a47fa9f47040b83f42ada9dfd053d47281ae74.

The following previously published dataset was used:

| Author(s) | Year | Dataset title | Dataset URL | Database and Identifier |
|---|---|---|---|---|
| Bakr S, Gevaert O, Echegaray S | 2018 | NSCLC Radiogenomics | https://wiki.cancerimagingarchive.net/display/Public/NSCLC+Radiogenomics | TCIA, Radiogenomics |

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

## Appendix 1

### Scanner parameter and follow-up strategies

CT scans ranged from thoracic inlet to subcostal plane and were obtained before surgical resection from two CT machines: Brilliance (Philips Medical Systems Inc, Cleveland, OH) and SOMATOM Definition AS (Siemens Aktiengesell-schaft, Munich, Germany).

CT parameters of Brilliance (Philips Medical Systems Inc) were as follows: $64 \times 1$ mm$^2$ acquisition; 0.75 s rotation time; slice width 1 mm; tube voltage, 120 kVp; tube current, 150–200 mA; lung window centre: –700 Hounsfield units (HU) and window width: 1200 HU; mediastinal window centre: 60 HU and window width: 450 HU level; pitch: 0.906; and field of view (FOV): 350 mm.

CT parameters of the SOMATOM Definition AS (Siemens Aktiengesell-schaft) were as follows: $128 \times 1$ mm$^2$ acquisition; 0.5 s rotation time; slice width: 1 mm; tube voltage: 120 kVp; tube current: 150–200 mA; lung window centre: –700 HU and window width: 1200 HU; mediastinal window centre: 60 HU and window width: 450 HU level; FOV: 300 mm; pitch: 1.2; and FOV: 350 mm. CT images were reconstructed into 0.67 to 1.25 mm section thicknesses according to a high-resolution algorithm.

Follow-up was conducted through outpatient examinations or telephone calls.

Chest CT scan and abdominal ultrasound/CT were performed on follow-up visits within a duration of 3, 6, and 12 months after operation and annually thereafter for 5 years. Magnetic resonance imaging for brain and bone scan were annually performed for 5 years or when the patient had signs or symptoms of recurrence.

### Similarity score definition

Similarity score for patient $x$ and patient $y$:

$$Sim\left(x,\ y\right) =\ C_{xy} * L_{xy} * H_{xy} * T_{xy}$$

$C_{xy}$ : if $x$ and $y$ have same gender, get 1 point; if $x$ and $y$'s age difference is within 5 year, get another 1 point.

$L_{xy}$ : if $x$ and $y$'s tumours locate at the same lung lobes, get 1 point.

$H_{xy}$ : if $x$ and $y$'s histology of tumours is the same type, get 1 point.

$T_{xy}$ : if $x$ and $y$ have the same T stage, get 1 point; if $x$ and $y$ have the same N stage, get another point; if $x$ and $y$ have the same M stage, get another 1 point.

When $Sim\left(x,\ y\right) > 0$, patient $x$ and $y$ can be connected.

### ResNet-Graph NRI and IDI results

Transformer-Graph comparing with ResNet-Graph, regarding NRI (OS: 0.240, 95% CI: –0.325 to 0.600, p<0.001; RFS: 0.104, 95% CI: –0.41 to 0.389, p<0.001) and IDI (OS: 0.075, 95% CI: 0.068–0.082, p<0.05; RFS: 0.063, 95% CI: 0.027–0.098, p<0.05).

