## [Editor Report]

This work constructed a population graph deep learning model using machine learning-generated imaging features and non-imaging clinical characteristics that were proven to be effective at predicting the survival of patients with early-stage NSCLC, which help us understand the imaging and non-imaging features in determining NSCLC populations with a high risk of recurrence, and the high predictive accuracy proves its novelty and significance.

---

## [Decision Letter]

**Decision letter after peer review:**

Thank you for submitting your article "Early-Stage NSCLC Patients' Prognostic Prediction with Multi-information Using Transformer and Graph Neural Network Model" for consideration by *eLife*. Your article has been reviewed by 2 peer reviewers, and the evaluation has been overseen by a Reviewing Editor and Caigang Liu as the Senior Editor. The reviewers have opted to remain anonymous.

Essential revisions:

1) Additional information regarding the external validation cohort requires to be supplemented, and the scale of the external independent cohort would better be enriched if possible;

2) The current situation of this field, along with the limitations of this study could be discussed in more detail in the Discussion.

*Reviewer #1 (Recommendations for the authors):*

While I think something like deep learning is needed in the medical field, I think the authors could do substantially more to validate that the Transformer-Graph model is the right choice and this work is appropriately placed in the larger context of ongoing research in the field. Although this approach is of great significance, the manuscript contains a number of weaknesses. In order to improve the current manuscript, I have the following critical points which need to be addressed before publication.

1) In the statistical analysis on page 8 line 180, I find it hard to understand "some code".

Please clarify this and add the missing part.

2) For Figures 3C and D, only KM curves on the test data set for OS and RFS prediction are presented. The author should also supply KM curves on external data sets for OS and RFS prediction if it is possible.

3) The authors stated this manuscript is the first work to demonstrate the feasibility of using Transformer in survival prediction using a graph data structure and exploratory analysis of the models' intuitions in an attempt to explain these state-of-the-art methods, but the authors still need to discuss other key papers and more explicitly state their contribution.

4) In the discussion, the authors do a very good job to highlight the advantages and limitations of their study, but the following limitations should be expanded:

a. The authors think that the Transformer-Graph model's performance on the external set was below that of the testing set, maybe because of the patients' different demographics (e.g. age, cancer staging, gender), datasets originate (e.g. ethnicity, treatment, follow-up strategies) and the manual initial step. Smoking history is well known to play an important role in the development of lung cancer. If possible, the non-imaging clinical data should include smoking habits.

b. On page 14, the authors showed the male percentage is 66.7% and 78.3% in the testing set and external set. It is not clear why this is inconsistent with the data in Table 1. Please the authors clarify this.

5) Only one external dataset is not enough to evaluate the Transformer-Graph model. If possible, the authors should analyze more datasets.

*Reviewer #2 (Recommendations for the authors):*

I have a few concerns that I believe need to be addressed before publication, but overall, I am enthusiastic about this work being in *eLife*.

1) The authors analyzed specifically as an example, patient No 44. 1 or 2 more examples could be helpful to understand the benefit of this model for survival prediction.

2) The model was constructed using both imaging and non-imaging clinical features. More discussion of the benefit of this model compared to most current prognostic prediction methods could be helpful.

3) If possible, increase the clinical case to decrease the different patients' demographics.

---

## [Author Response]

Essential revisions:1) Additional information regarding the external validation cohort requires to be supplemented, and the scale of the external independent cohort would better be enriched if possible;

We have added the KM survival plots for the external set as Figure 4—figure supplement 1. Other additional information regarding the external validation cohort has been added as Supplementary File 1.

We have endeavored to try and find another external dataset. We searched extensively over 40 lung cancer datasets on the TCIA and found only 2 of them met our requirements. The first dataset is already used as our current external dataset. The other dataset we found that the demographics, treatment and follow-up periods were vastly different. For example, the dataset has death rate of 95.88% for early-stage lung cancer patients which is very high, but this could be due to the fact that the dataset is very old (different treatment strategies, etc). We even proceeded to test a model with the second dataset, but the results were not satisfactory which we think is due to differences between the training cohorts and the testing cohorts. To this end, we feel that we are only able to include one external dataset. We have listed “only one external set” as one of our limitations.

2) The current situation of this field, along with the limitations of this study could be discussed in more detail in the Discussion.

Thank you for your suggestion. We have added and elaborated further with respect to the limitations of this study.

Reviewer #1 (Recommendations for the authors):While I think something like deep learning is needed in the medical field, I think the authors could do substantially more to validate that the Transformer-Graph model is the right choice and this work is appropriately placed in the larger context of ongoing research in the field. Although this approach is of great significance, the manuscript contains a number of weaknesses. In order to improve the current manuscript, I have the following critical points which need to be addressed before publication.1) In the statistical analysis on page 8 line 180, I find it hard to understand "some code".Please clarify this and add the missing part.

This is an error caused by Endnote citation generation, we have removed it.

2) For Figures 3C and D, only KM curves on the test data set for OS and RFS prediction are presented. The author should also supply KM curves on external data sets for OS and RFS prediction if it is possible.

We have added the KM curves on external data sets for OS and RFS prediction set as Figure 4—figure supplement 1. Thank you for your suggestion.

3) The authors stated this manuscript is the first work to demonstrate the feasibility of using Transformer in survival prediction using a graph data structure and exploratory analysis of the models' intuitions in an attempt to explain these state-of-the-art methods, but the authors still need to discuss other key papers and more explicitly state their contribution.

Thank you for your suggestion. We have further elaborated and summarized some related papers and explained our contribution in the Discussion section.

4) In the discussion, the authors do a very good job to highlight the advantages and limitations of their study, but the following limitations should be expanded:a. The authors think that the Transformer-Graph model's performance on the external set was below that of the testing set, maybe because of the patients' different demographics (e.g. age, cancer staging, gender), datasets originate (e.g. ethnicity, treatment, follow-up strategies) and the manual initial step. Smoking history is well known to play an important role in the development of lung cancer. If possible, the non-imaging clinical data should include smoking habits.

Thank you for your suggestion. It is unfortunate that we do not have the smoking history information in the main set. This is an inherent limitation of a retrospective study. We agree that this would have been important information to include. We have listed this in the limitation.

b. On page 14, the authors showed the male percentage is 66.7% and 78.3% in the testing set and external set. It is not clear why this is inconsistent with the data in Table 1. Please the authors clarify this.

Sorry, this was an erroneous typo. We have corrected it.

5) Only one external dataset is not enough to evaluate the Transformer-Graph model. If possible, the authors should analyze more datasets.

We have endeavored to try and find another external dataset. We searched extensively over 40 lung cancer datasets on the TCIA and found only 2 of them met our requirements. The first dataset is already used as our current external dataset. The other dataset we found that the demographics, treatment and follow-up periods were vastly different. For example, the dataset has death rate of 95.88% for early-stage lung cancer patients which is very high, but this could be due to the fact that the dataset is very old (different treatment strategies, etc). For more details pleaes refer to here:

(https://wiki.cancerimagingarchive.net/display/Public/NSCLC-Radiomics#1605685483aa3937478c4031873e85766dfdde48).

We even proceeded to test a model with the second dataset, but the results were not satisfactory which we think is due to differences between the training cohorts and the testing cohorts. To this end, we feel that we are only able to include one external dataset. We have listed “only one external set” as one of our limitations.

Reviewer #2 (Recommendations for the authors):I have a few concerns that I believe need to be addressed before publication, but overall, I am enthusiastic about this work being in eLife.1) The authors analyzed specifically as an example, patient No 44. 1 or 2 more examples could be helpful to understand the benefit of this model for survival prediction.

Thank you for your great suggestion. We have added another patient No.182 in the subanalysis part. As No 44 is a high-risk patient, we then analyzed a low-risk patient (No. 182) as a comparison to help understand the model.

2) The model was constructed using both imaging and non-imaging clinical features. More discussion of the benefit of this model compared to most current prognostic prediction methods could be helpful.

We have elaborated more in the discussion with respect to the TNM and the ResNet-Graph model.

We have now also made it more clear that the ResNet-Graph model used both imaging and non-imaging data in the previous version with additional information and comparison in the supplementary section.

3) If possible, increase the clinical case to decrease the different patients' demographics.

We have endeavored to try and find another external dataset. We searched extensively over 40 lung cancer datasets on the TCIA and found only 2 of them met our requirements. The first dataset is already used as our current external dataset. The other dataset we found that the demographics, treatment and follow-up periods were vastly different. For example, the dataset has death rate of 95.88% for early-stage lung cancer patients which is very high, but this could be due to the fact that the dataset is very old (different treatment strategies, etc). For more details pleaes refer to here:

(https://wiki.cancerimagingarchive.net/display/Public/NSCLC-Radiomics#1605685483aa3937478c4031873e85766dfdde48).

We even proceeded to test a model with the second dataset but the results were not satisfactory which we think is due to differences between the training cohorts and the testing cohorts. To this end, we feel that we are only able to include one external dataset. We have listed “only one external set” as one of our limitations.